# Bond Degradation Mechanism and Constitutive Relationship of Ribbed Steel Bars Embedded in Engineered Cementitious Composites under Cyclic Loading

**DOI:** 10.3390/ma16010252

**Published:** 2022-12-27

**Authors:** Jiaojiao Pan, Tingting Lu, Mingke Deng

**Affiliations:** 1School of Civil Engineering, Xijing University, Xi’an 710123, China; 2Shaanxi Key Laboratory of Safety and Durability of Concrete Structures, Xi’an 710123, China; 3School of Civil Engineering, Xi’an University of Architecture and Technology, Xi’an 710055, China

**Keywords:** engineered cementitious composites, cyclic loading, ribbed steel bar, bond degradation mechanism, constitutive relationship

## Abstract

In order to investigate the bond degradation mechanism and constitutive relationship of ribbed steel bars in engineered cementitious composites (ECCs) under cyclic loading, 12 groups of specimens were tested in this paper. The design parameters included ECC compressive strength, ECC flexural toughness, cover thickness, and anchorage length. The results indicated that the degradation of the bond behavior of the ribbed steel bars in the ECCs under cyclic loading was mainly caused by the degradation of the properties of the ECC material itself, concentrating on the development of cracks in the ECC, the extrusion and shear failure of the ECC between the steel bar ribs, and the continuous grinding of the ECC particles on the shear failure surface. The degradation of the bond stress–slip curves under cyclic loading was mainly reflected by the degradation in the ultimate bond strength and unloading stiffness. According to the monotonic loading test results, a monotonic bond stress–slip relationship model was proposed. On this basis, through building the hysteretic rules of the bond stress–slip curves under cyclic loading, a calculation model was proposed to predict the bond stress–slip constitutive relationship between the ribbed steel bars embedded in the ECCs under cyclic loading. Finally, the validity of the proposed model was verified by a comparison between the model curves and the tested curves.

## 1. Introduction

Engineered cementitious composites (ECCs) [1,2] represent a new kind of building material with high strength, high toughness, and high damage tolerance. They are characterized by an excellent strain hardening behavior with multiple fine cracks when subjected to tensile loading [3,4]. Due to these outstanding mechanical properties, ECCs have been widely utilized in new structures [5,6] and the repair of existing structures [7]. In these structures, bond behavior is the foundation of the combined action between the steel bars and ECC in various operating environments and stress conditions. Therefore, some studies have been carried out to evaluate the bond behavior of steel bars in ECCs under monotonic loading in recent years. Chao [8] investigated the effects of the type and volume content of the fibers regarding the bond behavior of the reinforcing bar in tensile strain–hardened fiber–reinforced cement composites via the pullout experiment. Bandelt [9] tested the beam specimens with lap splices in four–point bending tests to examine the influence of the concrete cover and steel confinement on the bond stress–slip behavior of steel bars embedded in high–performance fiber–reinforced cementitious composite (HPFRCC) materials. The results indicated that the tensile strength and strain capacity of the matrix were the two important factors that dominate bond performance. Due to the remarkable tensile strain hardening performance, good strain compatibility between steel bars and ECCs was achieved. Consequently, the bond behavior in reinforced ECC structures obviously improved, which induced a smaller splitting crack when compared to ordinary reinforced concrete specimens [10,11]. Lee [12] discussed the effect of embedded length on bond strength and then came up with the design suggestion of anchorage length for the reinforcing bars in the ECC members. Li [13] evaluated the bond performance of the reinforcing bars in the HPFRCCs before and after exposure to elevated temperatures.

The bond behavior of steel bars embedded in concrete under cyclic loading has an important influence on the restoring force characteristic of the structures, which is of great significance to the general seismic design and finite element analysis of structures [14,15]. Existing research [16,17] showed that bond degradation under cyclic loading was the main factor resulting in a loss of strength and stiffness in a joint zone. For ordinary reinforced concrete structures, much attention has been paid to the bond stress–slip behavior as a key issue in the assessment of seismic capacity. Morita [18] first proposed the bond stress–slip relationship model between steel bars and ordinary concrete under cyclic loading. This model used double–broken lines to simulate the ascending stage of a bond stress-slip curve under monotonic loading but did not take the descending stage of the bond stress–slip curve into account. The bond stress–slip relationship model established by Tassions [19] adopted the same bond stress–slip relationship curve for monotonic tension and compression, in which the influence of the number of cycles on bond degradation was taken into account. However, this model did not consider the degradation of the outer boundary of the bond stress–slip hysteretic curve under cyclic loading. Eligehausen [20] observed that for specimens with transverse stirrup restraint, the degree of bond degradation was controlled by the shear failure degree of the concrete between the steel bar ribs. A greater control slip would generate a larger shear failure range for the concrete between the steel bar ribs, leading to more serious bond degradation. Eligehausen proposed a bond stress–slip model for steel bars in ordinary concrete with transverse stirrups, which was suitable for low–cycle loadings with large slips and few cycles.

Due to fiber bridging stress, the ECC material could still carry a greater tensile load after cracking, which had effective control over the opening and extension of cracks [21]. The bond behavior of steel bars in ECCs under cyclic loading has been investigated in previous research [22]. The results suggested that the bond performance of the ribbed steel bars embedded in ECCs under cyclic loading could be improved due to the excellent damage tolerance ability of the ECC materials. There are significant differences in the bond degradation mechanism and bond stress–slip hysteretic curve between the ECC specimens and ordinary concrete specimens. However, there are few pieces of research on the bond stress–slip constitutive relationship between ribbed steel bars and ECCs under cyclic loading.

A total of 12 groups of pullout specimens were tested in this paper to evaluate the bond degradation mechanism and constitutive relationship of ribbed steel bars embedded within engineered cementitious composites under cyclic loading. Damage factors were introduced to reflect the influence of ECC damage development on the degradation of positive peak bond stress. Considering the influence of slip on the degradation laws of positive peak bond stress, unloading stiffness, horizontal friction resistance, and negative peak bond stress, a calculation model for the bond stress–slip constitutive relationship and hysteretic rules between the ribbed steel bars and ECCs under cyclic loading was proposed.

## 2. Experiment Scheme

### 2.1. Test Materials

ECC consisted mainly of Ordinary Portland Cement (P•O 42.5 R), fly−ash, mineral powder, silica fume, fine river sand, water, and polyvinyl alcohol (PVA) or polyethylene (PE) fibers. This paper prepares six ECC mixtures for the purpose of investigating the effects of ECC compressive strength and flexural toughness on bond behavior. The designed proportions and mechanical properties of the ECC mixtures are listed in Table 1, where *f*_cu_ is the cubic compressive strength, *T* is the flexural toughness, and *f*_t_ is the tensile strength. The flexural toughness can be calculated referring to the paper [22].

### 2.2. Specimen Preparation

Pullout specimens are used in this research, as shown in Figure 1. Twelve groups of specimens were prepared, and each group included one for monotonic loading and one for cyclic loading. Table 2 shows the design details of all the specimens. Design parameters included ECC compressive strength, ECC flexural toughness, cover thickness, and anchorage length. As shown in Table 2, letters *c*, *d*, and *l*_a_ represent the cover thickness, bar diameter, and embedded length, respectively.

### 2.3. Test Setup and Instrumentation

Figure 2 illustrates the test setup used in this paper, where a MTS 250 kN test machine supplied the applied load. The displacement control mode was used with a constant rate of 1 mm/min for the monotonic loading. The control displacements for cyclic loading were ±0.25, ±0.5, ±0.75, ±1, ±1.25, ±1.5, ±1.75, ±2, ±2.25, ±2.5, ±2.75, ±3, ±3.5, ±4, ±4.5, ±5, ±5.5,±6, and ±7 mm. The loading rate was 1 mm/min when the control displacement was less than 3 mm, and, following that, a rate of 2 mm/min was used. Two linear variable differential transducers (LVDTs) were installed at the free end to monitor the change of relative slip between the steel bar and the ECCs.

## 3. Bond Degradation Mechanism Analysis

Typical bond stress–slip curves of the specimens under monotonic and cyclic loading are shown in Figure 3. Test results of the specimens under monotonic and cyclic loading are summarized in Table 2.

As shown in Table 2, $\tau_{u}^{+}$ and $\tau_{u}^{-}$ are the positive and negative peak bond stresses under cyclic loading, respectively. $\tau_{r}^{+}$ and $\tau_{r}^{-}$ are the positive and negative residual bond stresses under cyclic loading, respectively. $\tau_{u}^{m}$ and $\tau_{r}^{m}$ are the ultimate bond strength and residual bond strength under cyclic loading, respectively ($\tau_{u}^{m}=(\tau_{u}^{+}+\tau_{u}^{-})/2$, $\tau_{r}^{m}=(\tau_{r}^{+}+\tau_{r}^{-})/2$). $\tau_{u}$ and $\tau_{r}$ are the ultimate bond strength and residual bond strength under monotonic loading, respectively. *K*_r_ is the ratio of residual bond strength to ultimate bond strength under monotonic loading. $\lambda_{u}$ is the degradation coefficient of ultimate bond strength under cyclic loading.

The ultimate bond strength and residual bond strength were two important turning points of the bond stress−slip curve, and the corresponding slips of these two points were marked as *s*_u_ and *s*_r,_ respectively. Based on these two characteristic slips, the development process of the bond stress−slip curve was divided into three stages. According to the characteristics of experimental curves, Figure 4 depicts the simplified diagram of a bond stress–slip curve at a given loading cycle.

Then, the bond degradation mechanism between the ribbed steel bar and the ECC under cyclic loading was analyzed when the controlled displacement, *s*_con_, was under different stages, as follows.

Stage 1:

Figure 4 shows the schematic diagram of the position of the steel bar rib and the damage development of ECC at a given cycle. As seen in Figure 4, the complete loading process consists of the following six stages: ascending stage (A–B) in the positive direction, descending stage (B–C) in the positive direction, horizontal stage (C–D), ascending stage (D–E) in the negative direction, descending stage (E–F) in the negative direction, and the horizontal stage (F–G) in the opposite direction.

(1) Ascending stage (A–B) in the positive direction

Point A is the unloaded state. When positively loading the control displacement, the bond stress–slip curve reached a peak at point B. The steel bar rib moved to the right, extruding the right ECC, and was separated from the left ECC at the same time. A schematic diagram of the position of the steel bar rib is shown in point B in Figure 5. The bond mechanism in this stage is the same as that in the monotonic loading ascending stage. At the initial loading stage, the bond resistance was governed by the chemical adhesion between the steel bar and the cement gels. The bond stress–slip curve increased linearly, and the specimen was in the elastic stage. With an increase in the pullout load, the steel bar at the loaded end began to slip, and the chemical adhesion in this part was destroyed. Then, the bond resistance was dependent on the mechanical interlocking and friction between the ribs and the surrounding ECC. In this stage, the slip was mainly caused by the local extrusion deformation of the matrix at the root of the steel bar ribs. When the slip occurred at the free end, the chemical adhesion in the whole bond region was completely lost.

The crack development in the ECC during the pullout process of the steel bar is shown in Figure 6. During the pullout process of the steel bar, internal inclined cracks came into being due to the oblique extrusion force of the steel bar ribs on the ECC. The ECC around the steel bar was cut into a series of inclined compression cones, the top of which was subjected to the combined action of the extrusion force and friction force. Due to the fiber bridging stress across the cracking section effectively preventing the cracks from further widening, the extrusion force was redistributed to the whole matrix. As the pullout load continued to increase, the ECC in front of the steel bar rib was gradually crushed. The crushed ECC powder compactly piled up at the root of the steel bar rib. At the same time, an interval area was generated behind the steel bar rib. Finally, a new slip plane formed with an inclination angle generally between 30 and 40 degrees [23] (see Figure 6c).

According to the thick–wall cylinder theory of elastic mechanics [24], the radial component of extrusion stress would make the surrounding ECC around the steel bar to be under circumferential tension. The distribution of the circumferential tensile stress of an ECC is shown in Figure 6b. As seen in Figure 6b, the farther away from the contact interface between the steel bar and ECC, the smaller the circumferential tensile stress is. When the circumferential tensile stress was in excess of the initial cracking strength of the matrix, the internal radial crack was induced perpendicular to the direction of the steel bar. The circumferential tensile stress at the interface between the reinforcing bar and the ECC was the largest. Therefore, the radial crack first appeared near this interface. Besides, it should be mentioned that the internal radial crack developed rather slowly for the ECC specimens because of the constraint effect of the fiber bridging stress.

(2) Descending stage (B–C) in the positive direction

Once control displacement was reached, the unloading was underway. When the external force dropped to zero (point C), the elastic deformation of the ECC matrix recovered entirely, and the internal inclined cracks were closed. By reason of the nonreversible plastic deformation that resulted from the crushed ECC, the left interval after unloading saw a slight reduction but still existed, with the width approximately equal to the residual slip at point C.

(3) Horizontal stage (C–D).

Then, the steel bar ribs started to move to the left as reverse loading began. In this stage, the bond resistance was quite weak, which mainly relied on the interface friction between the steel bar and the surrounding matrix. As the value of the slip decreased rapidly, the bond stress basically remained constant, and the bond stiffness approached zero, consequently forming the horizontal stage of the bond stress−slip curve. At the point of D, the steel bar rib contacted the matrix again in the opposite direction, and a new interval was observed to the right of the steel rib.

(4) Ascending stage (D–E) in the negative direction.

As the slip was reduced to zero, the steel bar rib was in complete contact with the left ECC. Then, a rapid increase was found both in the bond stress and bond stiffness with the increase in the reverse load. The bond characteristic of the D–E stage was similar to that of the A–B stage. Finally, an internal inclined crack occurred in the left ECC (point E) due to the extrusion of the rib. Closure of the crack in the right side of the steel bar rib was observed, but a larger interval was left.

(5) Descending stage (E–F) in the negative direction.

The feature of curve at the E–F stage was similar to that of the B–C stage. Bond stress saw an obvious reduction by reason of the reversed load being removed. However, the residual slip could not be recovered absolutely. This phenomenon was caused by the nonreversible plastic deformation and the local damage induced by the matrix crushing. At point F, positive loading was applied again.

(6) Horizontal stage (F–G) in the opposite direction.

Similar to the features of the C–D stage, the bond stress saw negligible fluctuation, and the bond stiffness was close to zero at this stage, which forms the reverse horizontal stage of the bond stress−slip curve. At point G, the steel bar rib fully contacted, with the matrix once more in the positive direction, and then a complete loading cycle was finished.

Under the condition of small control displacement (i.e.,$s_{\mathrm{son}}<s_{u}$), inclined cracks and radial cracks appeared in the matrix, but their extension length was relatively small, and they were still in a stable development stage. Besides, although the crushed ECC powder had been densely gathered in front of the transverse rib of the steel bar, the shear compression failure range of the ECC was small at this time and was only confined to the local area near the transverse rib of the steel bar. The damage degree of the ECC around the steel bar was relatively slight. In this stage, the bond stress was mainly dependent on the interlocking action and the friction action of the interface between the steel bar and the ECC.

Stage 2:

In this stage, the specimen quickly reached peak bond stress as the displacement increased to the control value. The damage degree of the ECC was severe, for which the shear compression failure range was close to half that of the distance between the transverse ribs of the reinforcing bar. In the process of reverse loading, the extrusion bearing capacity of the ECC between the steel bar ribs was greatly reduced because its shear compression damage range had been large. Thus, both the bond strength and stiffness decreased during the reverse loading of the controlled displacement.

When the radial crack intersected with the inclined crack, the dentate cone was divided into isolated and almost free of the lateral constraint outriggers. The bending strength and shear strength of these outriggers were greatly reduced, which resulted in the formation of a softening boundary layer around the steel bar. As a result, the holding effect of the ECC over the steel bar was weakened. At this point, bond resistance was mainly provided by the mechanical interlocking between the transverse rib of the steel bar and the undamaged ECC. The friction effect on the contact interface between the steel bar rib and the ECC was partially transformed into the friction−interlocking effect between the ECC that had been cut off from the shear compression failure surface.

In the case of greater control displacement (i.e.,$s_{u}\leq s_{\mathrm{son}}\leq s_{r}$), after the positive and negative loading cycles were completed, the shear compression failure range generated by them would be connected, greatly weakening the extrusion bearing capacity of the ECC between the steel bar ribs. In the next cycle, bond resistance consisted of the mechanical interlocking between the steel bar rib and the undamaged ECC, as well as the frictional action between the ECC particles on the shear compression failure surface. With an increase in control displacement, the ECC between the steel bar ribs was crushed, and finally, the softening boundary layer around the steel bar was gradually flattened. As a result, peak bond stress degraded continuously with the increase in control displacement (see Figure 6b).

Stage 3:

When the control displacement, *s*_con_, was more than the residual slip, *s*_r_, the slip of the steel bar was approximately half that of the rib spacing as the displacement increased to the control value. The ECCs between the steel bar ribs were almost completely cut off, leading to the mechanical interlocking vanishing completely. Then, bond resistance was mainly dependent on the frictional interlocking action between the ECC particles on the failure surface of the shear compression. Bond stress, at this stage, was close to the residual bond strength under monotonic loading. With the increase in control displacement, the ECC particles on the shear failure surface gradually became fine, and finally, the interface friction remained constant. As a result, the degradation of the bond behavior tended to become stable.

Based on the above analysis of the bond mechanism, it can be concluded that the degradation of the bond behavior under cyclic loading was essentially caused by the degradation of the properties of the ECC materials, which were involved in the development of cracks in the ECCs, the extrusion and shear failure of the ECCs between the steel bar ribs, and the continuous grinding of the ECC particles on the shear failure surface. With respect to the different control displacements, the cracking degree of the ECCs and the shear compression failure degree of ECCs between the ribs were different. Thus, the degradation rate and degradation degree of bond behavior were also different.

The degradation of the bond performance under cyclic loading was primarily reflected in the degradation of the ultimate bond strength and the degradation of the unloading stiffness. On the one hand, under the action of high local stress, the ECC between steel bar ribs would produce different degrees of microextrusion and microcracking. These deformations continually increased with the alternate actions of the tensile and compressive loads, consequently leading to the degradation of the unloading stiffness. On the other hand, the plastic deformation of the ECC caused by extrusion and cracking cannot be recovered after unloading and accumulated under the cyclic action of tensile and compressive loads, which resulted in a continuous increase in residual slip when unloading to zero. These two factors together led to the degradation of the ultimate bond strength.

## 4. Degradation of Ultimate Bond Strength

The ratio of ultimate bond strength under cyclic loading to that under monotonic loading is defined as the degradation coefficient of ultimate bond strength: $\lambda_{u}$. It reflects the degradation degree of the ultimate bond strength under cyclic loading. A smaller degradation coefficient means more severe degradation. The influence of various parameters on the degradation coefficient of the ultimate bond strength is shown in Figure 7.

(a) ECC compressive strength

With an increase in ECC compressive strength, the degradation coefficient of the ultimate bond strength increased first and then decreased slightly (see Figure 7a). This phenomenon may be caused by greater ECC compressive strength yielding an excellent damage resistance in the ECC matrix, consequently reducing the degradation degree of the ultimate bond strength. However, with a further increase in ECC compressive strength, the pullout force applied on the steel bar is greater, which led to the quick development of cracks and the severe crush of the ECC between the steel bar ribs. The damage to the ECC continued to accumulate under the cyclic action of positive and negative loading. As a result, the degradation degree of the ultimate bond strength was larger.

(b) ECC flexural toughness

From Figure 7b, it can be seen that the degradation coefficients of the ultimate bond strength for specimens dB1, dB2, and dB3 have extremely small differences. It indicates that the increase in ECC toughness had not further benefited improvements to the degradation of the bond behavior. The degradation coefficients of the ultimate bond strength for these three group specimens were very close to 0.9, significantly higher than that of the other specimens. This phenomenon can be attributed to the fact that the addition of silica fumes not only filled the pores between the cement particles but also generated a gel with the hydration products, which increased the compactness and, thus, improved the damage tolerance ability of the ECC. As a result, the degradation of the bond behavior under cyclic loading was significantly reduced.

(c) Cover thickness

The enhancement of the cover thickness could increase the constraint effect of the ECC matrix on the cracks and control the extension of the splitting cracks, which reduces the damage to the specimens. Thus, the degradation degree of the ultimate bond strength decreased with an increase in cover thickness. From Figure 7c, it should also be noticed that the improvement effect is relatively small, although increasing cover thickness could improve the degradation of the ultimate bond strength. This was due to the development of cracks that had been effectively restricted by the fiber–bridging effect in the ECC.

(d) Embedded length

With an increase in embedded length, the degradation coefficient of the ultimate bond strength had a slight increase at first and then showed a dramatic reduction. As shown in Figure 7d, the degradation coefficient of the ultimate bond strength for specimen dE2 with an embedded length of 4d is the largest. From this phenomenon, it can be reasonably inferred that the degradation degree of the ultimate bond strength decreased with an increase in embedded length under the condition of an embedded length of less than 4d. However, when the embedded length exceeded 4d, the degradation degree of ultimate bond strength increased as the embedded length further increased.

## 5. Bond Stress–Slip Constitutive Relationship

### 5.1. Bond Stress–Slip Constitutive Relationship under Monotonic Loading

The bond stress−slip curve of the ribbed steel bar in the ECCs under monotonic loading consisted of an ascending stage, descending stage, and horizontal stage, which can be described by four characteristic points: the ultimate bond strength $\tau_{u}$ and its corresponding slip $s_{u}$, the residual bond strength $\tau_{r}$ and its corresponding slip $s_{r}$. A three–step model, as shown in Figure 8, is adopted to describe the bond stress–slip constitutive relationship between the ribbed steel bar and the ECCs.

On the basis of the least square method, the pull–out test results of the ribbed steel bars and the ECCs under a monotonic load were fitted to obtain the monotonic bond stress–slip constitutive model between the ribbed steel bars and the ECCs, which can be expressed as
(1)$$\tau =\left\{\begin{array}{cc} \tau_{u}{\left(s/s_{u}\right)}^{\chi } & s\leq s_{u}\\ \tau_{r}+\left[\left(s_{r}-s)(\tau_{u}-\tau_{r})/(s_{r}-s_{u}\right)\right] & s_{r}<s<s_{u}\\ \tau_{r} & s>s_{r} \end{array}\right.$$
where χ is the shape parameter, which represents the nonlinear characteristics of the ascending stage of the curve. Based on the least square method, the shape parameter χ was obtained as 0.13 (by the regression analysis of test results under monotonic load). According to the statistical analysis of the test results, $s_{u}$ was taken as 0.5 mm, and $s_{r}$ was taken as 7.5 mm.

The effects of ECC compressive strength, toughness, cover thickness, and embedded length on the ultimate bond strength were taken into account in this paper. According to the regression analysis of the test data, the calculation formula for the ultimate bond strength between the ribbed steel bar and the ECC under monotonic loading is described as
(2)$$\tau_{u}=2.5\times 10^{-5}\left(0.006T+0.802\right)(0.32+2.76\frac{c}{d})(0.72+1.12\frac{d}{l_{a}})f_{\mathrm{cu}}^{2.7}$$
where *c*/*d* = 3.41, if *c*/*d* ≥ 3.41 [21]; *l*_a_/*d* ≥ 4, if *l*_a_/*d* = 4; for PVA fiber specimens, *T* = 60, if *T* ≥ 60; for PE fiber specimens, *T* = 150, if *T* ≥ 150.

The correlation coefficient between the calculated and tested values of the ultimate bond strength was 0.973. Figure 9 shows a comparison between the tested and calculated values for ultimate bond strength. As shown in Figure 9, the calculated values fit well with the tested values.

Besides, the parameter *k*_r_ was marked as the ratio of the residual bond strength to the ultimate bond strength, which was $k_{r}=\tau_{r}/\tau_{u}$. Through the statistical analysis of the experimental results, the mean value of parameter *k*_r_ was 0.323, with a standard deviation of 0.092. Therefore, for ribbed steel bars:(3)$$\tau_{r}=0.323\tau_{u}$$

### 5.2. Hysteretic Rule of Bond Stress–Slip Curvse under Cyclic Loading

The bond stress between the steel bar and the ECC demonstrated an obvious degradation phenomenon when subjected to cyclic loading. The degradation degree of the bond stress under cyclic loading was mainly related to the number of cycles and the slip in the loading process [20]. In order to obtain the bond stress–slip hysteretic constitutive relationship under cyclic loading, a specific rule was usually introduced to reflect the hysteretic characteristics of the curves under cyclic loading on the basis of the bond stress–slip constitutive relationship under monotonic loading [25]. Thus, four characteristic parameters were introduced in this paper to develop the bond stress–slip constitutive model for ribbed steel bars embedded in ECCs under cyclic loading. These parameters included the positive peak bond stress $\tau_{s}^{+}$, unloading stiffness *K*, horizontal friction resistance $\tau_{f}$ and the negative peak bond stress $\tau_{s}^{-}$. A schematic diagram of the bond stress–slip relationship under cyclic loading is shown in Figure 10. According to the test results in Table 2, it was noticed that the ultimate bond strength degradation degree for the dA1, dD1, and dE3 specimens was very close. Therefore, these three groups of specimens were analyzed as the benchmark to investigate the degradation law of bond behavior under cyclic loading.

(1) Degradation of positive peak bond stress.

The damage factor, *d*_u_, was introduced to reflect the damage development process of ECC materials under cyclic loading, which led to the degradation of the positive peak bond stress. When loading to the same slip value, the relationship between the positive peak bond stress under cyclic loading and the bond stress under monotonic loading can be expressed as follows:(4)$$\frac{\tau_{s}^{+}}{\tau_{m}}=1-d_{u}$$
where $\tau_{s}^{+}$ is the positive peak bond stress in a certain loading cycle under cyclic loading; $\tau_{m}$ is the bond stress when monotonically loaded to the same slip *s*; $\tau_{m}$ can be calculated by Equation (1); *d*_u_ is the damage factor.

According to Equation (5), the damage factor, *d*_u_, can be expressed as
(5)$$d_{u}=1-\frac{\tau_{s}^{+}}{\tau_{m}}$$

The relationship between the damage factor and slip is shown in Figure 11.

As seen in Figure 11, the damage to the ECC materials continually increased with an increase in slip. The relationship between these two can be described by the following formula:(6)$$d_{u}=1-\exp(-\frac{s}{m})$$
where *s* is the slip, and m is the curve shape parameter.

Parameter m determines the slope of the rising stage of the curve. The smaller the value is, the higher the slope of the curve is. As a result, the faster the damage factor increases. Through the regression analysis of the test data, the parameter m was equal to 4.313.

Based on the analysis of the degradation law of the bond performance between the ribbed steel bars and the ECCs in Section 4, it can be noted that the influence of ECC compressive strength, flexural toughness, cover thickness, and embedded length on the bond performance under cyclic loading was mainly reflected by the degradation degree of the ultimate bond strength (i.e., $\lambda_{u}$). The degradation factor for ultimate bond strength $\lambda_{u}$ is the ratio of the ultimate bond strength under cyclic loading to that under monotonic loading. The greater the value of $\lambda_{u}$, the smaller the degradation degree of the ultimate bond strength under cyclic loading, the slower the damage development in the specimen, and the larger the value of parameter m in the damage factor calculation formula. Referring to Table 2, the mean value of the degradation coefficient for ultimate bond strength is 0.65. Herein, the parameter *m* is modified by the degradation coefficient for ultimate bond strength, and so
(7)$$m=4.313\frac{\lambda_{u}}{0.65}=6.635\lambda_{u}$$

(2) Degradation of unloading stiffness, *K*.

At the unloading stage, the bond stress–slip curve changed linearly. Figure 12 shows the relationship between unloading stiffness and slip.

As shown in Figure 12, the unloading stiffness degrades rapidly with an increase in slip at the initial loading stage. After the ultimate bond strength is reached, the degradation of the unloading stiffness is slow as the slip increases. The relationship between unloading stiffness, *K*, and slip, *s*, can be expressed as
(8)K=−24.6lns+126.1

The unloading stiffness was mainly related to the slip in the loading process. The influence of ECC compressive strength, flexural toughness, cover thickness, and embedded length on the unloading stiffness was very small and could be ignored.

(3) Degradation of the horizontal friction resistance

According to the bond mechanism analysis of the horizontal friction stage, it can be understood that the bond in this stage was dependent on the frictional resistance between the steel bar and the surrounding matrix. In a given loading cycle, there was a proportional relationship between the frictional resistance and peak bond stress. The ratio of the friction resistance to peak bond stress was close to a constant value. Therefore
(9)$$\frac{\tau_{f}}{\tau_{s}^{+}}=c$$

Through the statistical analysis of the test data, the mean value of parameter *c* was 0.050, with a standard deviation of 0.012. Thus,
(10)$$\tau_{f}=0.05\tau_{s}^{+}$$

(4) Degradation of the negative peak bond stress $\tau_{s}^{-}$

After positive loading was finished, the specimen was damaged, which resulted in the degradation of the negative peak bond stress. Therefore, it is necessary to consider the degradation of the negative peak bond stress in the same loading cycle. The ratio of negative peak bond stress to positive peak bond stress is shown in Table 3.

Its mean value was 0.825, with a standard deviation of 0.104. Thus,
(11)$$\tau_{s}^{-}=-0.825\tau_{s}^{+}$$

### 5.3. Bond Stress–Slip Constitutive Model under Cyclic Loading

For the loading cycle, *i*, it was supposed that there would be a maximum positive slip $s_{i,\max}^{+}$ and a maximum negative slip $s_{i,\max}^{-}$. Figure 13 demonstrates the calculation process. Then, the calculation model of the bond stress–slip hysteretic curve can be expressed as follows.

(1) Ascending stage in the positive direction: Peak point ($s_{i,\max}^{+}$,).

The ascending stage function was similar to that of the bond stress−slip constitutive model under monotonic loading. The calculation equation was
(12)$$\tau =\tau_{\mathrm{si},\max}^{+}{\left(\frac{s}{s_{i,\max}^{+}}\right)}^{\alpha }$$
where $\tau_{\mathrm{si},\max}^{+}$ is calculated by Equation (4), namely $\tau_{\mathrm{si},\max}^{+}=\tau_{m}(1-d_{u})$; $d_{u}$ is calculated by Equation (6); $\tau_{m}$ is the bond stress when monotonically loaded to the same slip *s*; $\tau_{m}$ is calculated by Equation (1).

The shape of the ascending curve depends on the value of the parameter *α*. When *α* is less than 1, the curve is convex. When *α* is equal to 1, the shape of the curve is linear. When *α* is greater than 1, the shape of the curve is concave. According to the statistical analysis of the test results, the value of parameter *α* is taken as 2.7 when the slip exceeds the corresponding slip of the ultimate bond strength under cyclic loading. If not, the value of parameter *α* is taken as 0.13.

(2) Descending stage in the positive direction: Peak point ($s_{i,\max}^{+}$,), unloading stiffness *K*.
(13)$$\tau =\tau_{\mathrm{si},\max}^{+}+K(s-s_{i,\max}^{+})$$
where *K* is calculated by Equation (8).

(3) Horizontal stage in the positive direction.

The bond stress in the horizontal stage was provided by the interface friction and basically remained constant. Thus,
(14)$$\tau =-\tau_{f}$$
where $\tau_{f}$ is calculated by Equation (10).

(4) Ascending stage in the negative direction: Peak point ($s_{i,\max}^{-}$,)

In the second half of the loading cycle, the degradation of the negative peak bond stress needed to be taken into account. The expression of the ascending stage in the negative direction is
(15)$$\tau =\tau_{\mathrm{si},\max}^{-}{\left(\frac{s}{s_{i,\max}^{-}}\right)}^{\alpha }$$
where $\tau_{\mathrm{si},\max}^{-}$ is calculated by Equation (11). The value of parameter *α* refers to Equation (12).

(5) Descending stage in the negative direction: Peak point ($s_{i,\max}^{-}$,$\tau_{\mathrm{si},\max}^{-}$), unloading stiffness *K*.

By observing the characteristics of the bond stress−slip curve between ribbed steel bar and ECC under cyclic loading, it was found that the change rule of the curve in the second half of the loading cycle was basically same as that in the first half of the loading cycle. Therefore, the unloading stage and the horizontal stage in the second half of the loading cycle can adopt the same calculation model as those in the first half of the loading cycle. That was
(16)$$\tau =\tau_{\mathrm{si},\max}^{-}+K(s-s_{i,\max}^{-})$$
where $\tau_{\mathrm{si},\max}^{-}$ is calculated by Equation (11); *K* is calculated by Equation (8).

(6) Horizontal stage in the negative direction

The bond stress in the horizontal stage was calculated by the following equation.
(17)$$\tau =\tau_{f}$$
where $\tau_{f}$ is calculated by Equation (10).

Any of the six loading cycles was selected to verify the accuracy of the calculation model. A comparison between the model curves and the experimental curves under cyclic loading is shown in Figure 14.

The model curves agreed well with the experimental curves. This indicated that the constitutive model proposed in this paper could effectively simulate the hysteretic characteristics and change rules of the bond stress−slip curves of ribbed steel bars in ECCs under cyclic loading.

## 6. Conclusions

This paper investigated the bond degradation mechanism and constitutive relationship of ribbed steel bars embedded in ECCs under cyclic loading based on previous experimental results. The conclusions are drawn as follows.

(1) The degradation of bond performance under cyclic loading was primarily reflected in the degradation of the ultimate bond strength and the degradation of the unloading stiffness. With regard to the different control displacements, the cracking degree of the ECC and the shear compression failure degree of the ECC between the ribs were different. The degradation rate and degradation degree for bond behavior were also different.

(2) With an increase in ECC compressive strength, the degradation degree of the ultimate bond strength decreased at first and then increased slightly. The degradation coefficients for the ultimate bond strength of specimens dB1, dB2, and dB3 were very close to 0.9, meaning a minor bond behavior degradation. This was due to the fact that the addition of silica fumes increased the compactness and thus improved the damage tolerance ability of the ECC.

(3) The bond stress–slip curve of the ribbed steel bars in the ECCs under monotonic loading consisted of an ascending stage, a descending stage, and a horizontal stage. Based on the tested results, the calculation formula for ultimate bond strength was built, and the monotonic bond stress–slip constitutive model was proposed.

According to the features of the bond stress–slip curve under cyclic loading, hysteretic rules were built. Considering the effect of slip on the characteristic parameters, namely, positive peak bond stress, unloading stiffness, horizontal friction resistance, and negative peak bond stress, a calculation model for the bond stress–slip constitutive relationship under cyclic loading was proposed. The model curves were in good agreement with the experimental curves, suggesting that the proposed calculation model in this paper is suitable for predicting the bond response of steel bars in ECCs under cyclic loading. Besides, based on the consideration of a limited number of tested specimens, the application range of the model proposed in this paper is limited to some extent.

## Figures and Tables

**Figure 1 materials-16-00252-f001:**
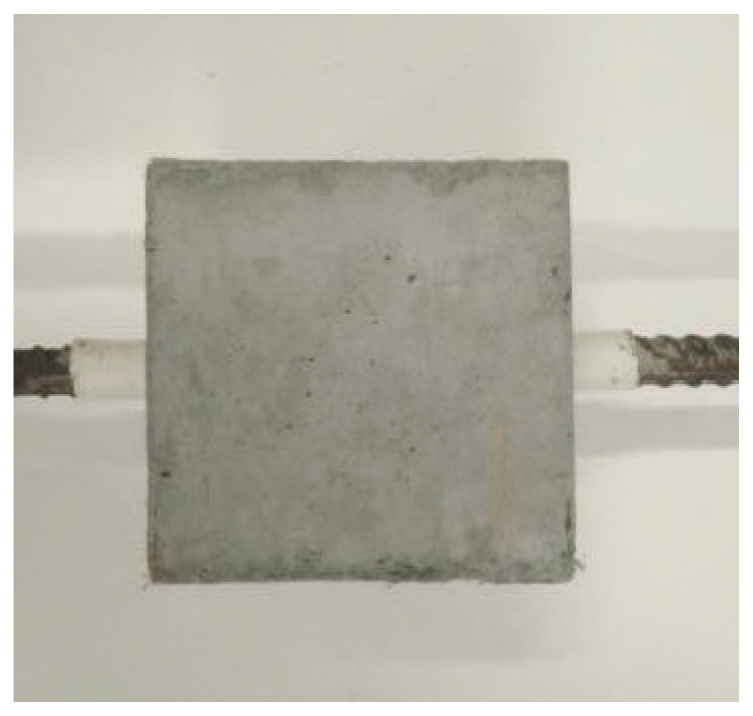
Tested specimens.

**Figure 2 materials-16-00252-f002:**
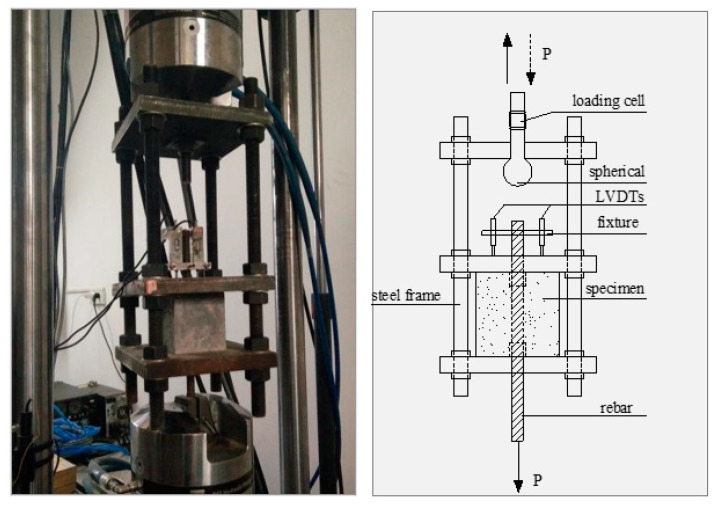
Test setup.

**Figure 3 materials-16-00252-f003:**
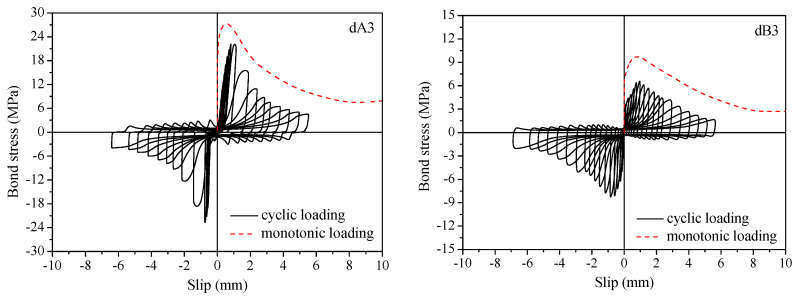
Typical bond stresss–lip curves of the specimens under monotonic and cyclic loading.

**Figure 4 materials-16-00252-f004:**
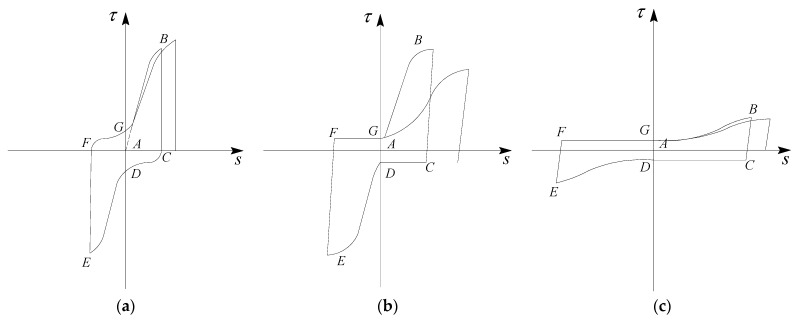
Simplified diagram of the bond stress−slip curve at a given loading cycle. (**a**) $s_{\mathrm{son}}<s_{u}$; (**b**) $s_{u}\leq s_{\mathrm{son}}\leq s_{r}$; (**c**) $s_{\mathrm{son}}>s_{r}$.

**Figure 5 materials-16-00252-f005:**
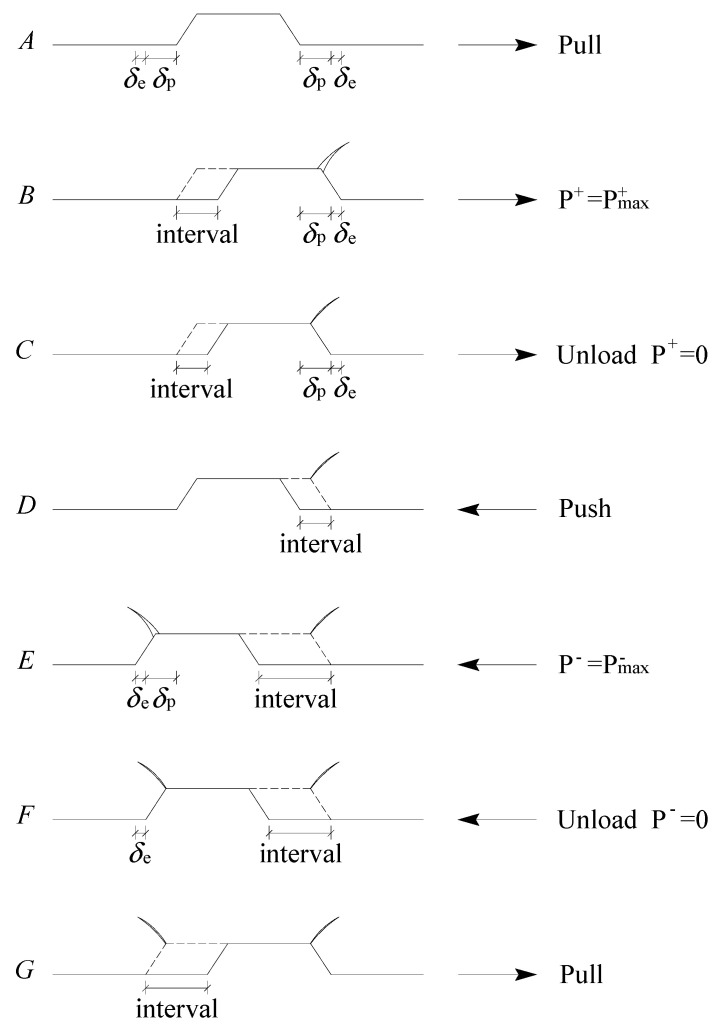
Schematic diagram of the position of the steel bar rib at a given cycle. $\delta_{e}$ and $\delta_{p}$ are the elastic and plastic deformation, respectively, in the loading process.

**Figure 6 materials-16-00252-f006:**
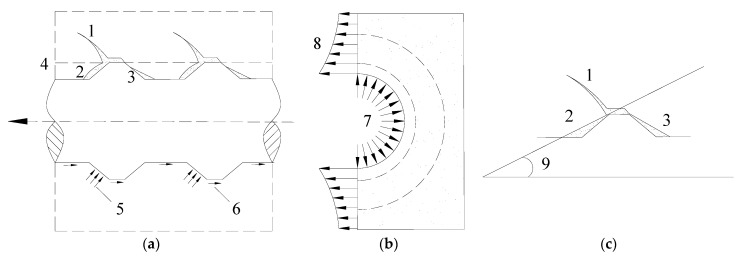
Crack development in the ECC during the pullout process of the steel bar. (**a**) Crack development in the ECC; (**b**) circumferential tensile stress in the ECC; (**c**) new slip plane. 1—internal inclined crack; 2—crushed area in front of the steel bar rib; 3—interval area behind the steel bar rib; 4—shear surface; 5—extrusion force; 6—friction; 7—radial extrusion stress; 8—circumferential tensile stress; 9—inclination angle.

**Figure 7 materials-16-00252-f007:**
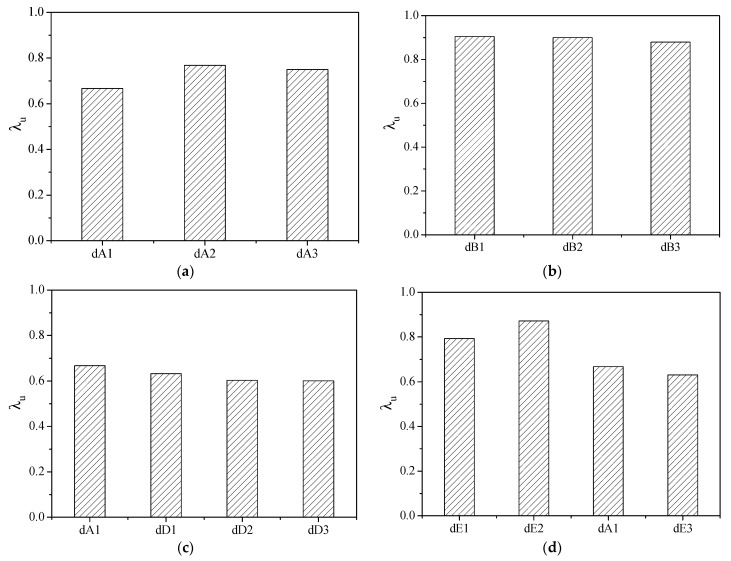
Influence of various parameters on the degradation coefficient of the ultimate bond strength. (**a**) compressive strength; (**b**) flexural toughness; (**c**) cover thickness; (**d**) anchorage length.

**Figure 8 materials-16-00252-f008:**
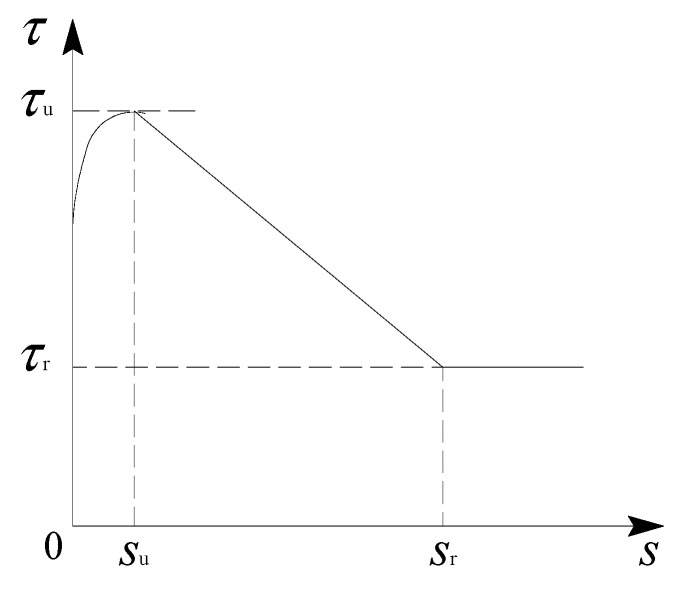
Schematic diagram of the bond−slip constitutive model under monotonic loading.

**Figure 9 materials-16-00252-f009:**
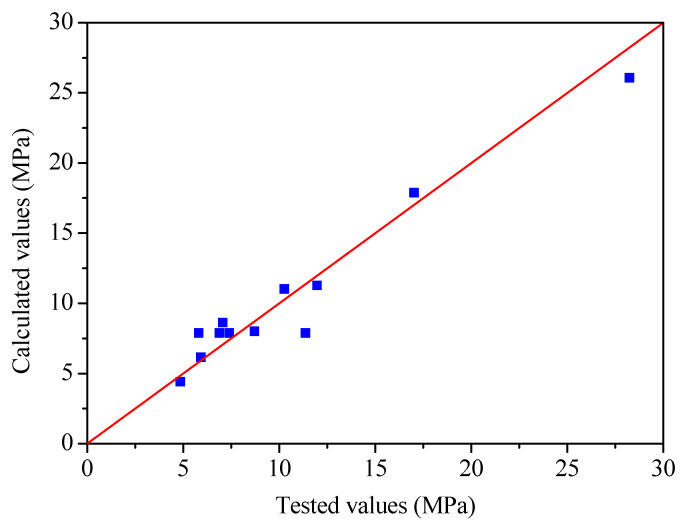
Comparison between the tested and calculated values for ultimate bond strength.

**Figure 10 materials-16-00252-f010:**
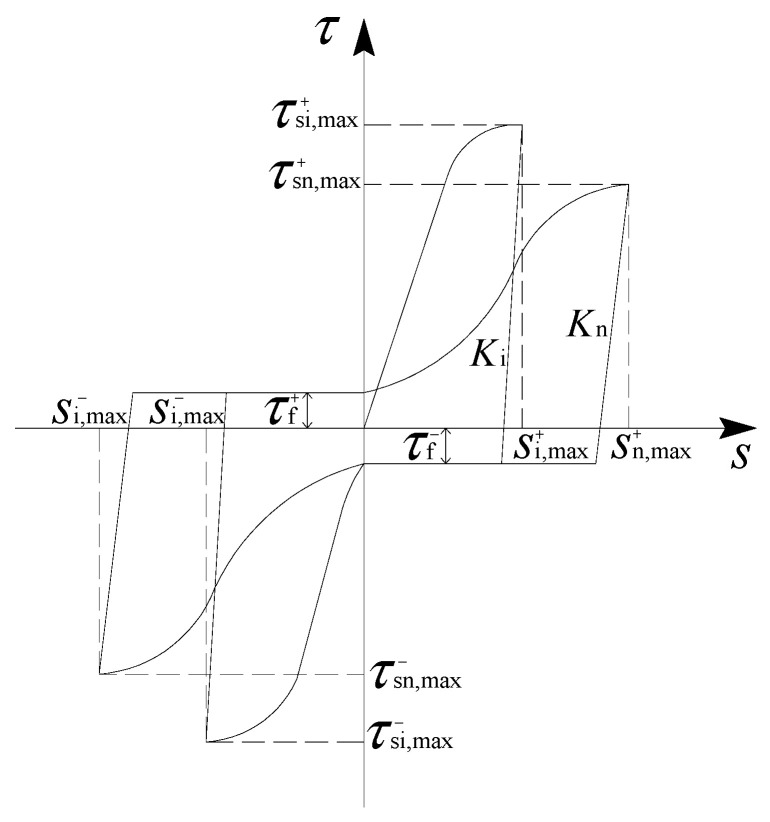
A schematic diagram of the bond–slip relationship under cyclic loading.

**Figure 11 materials-16-00252-f011:**
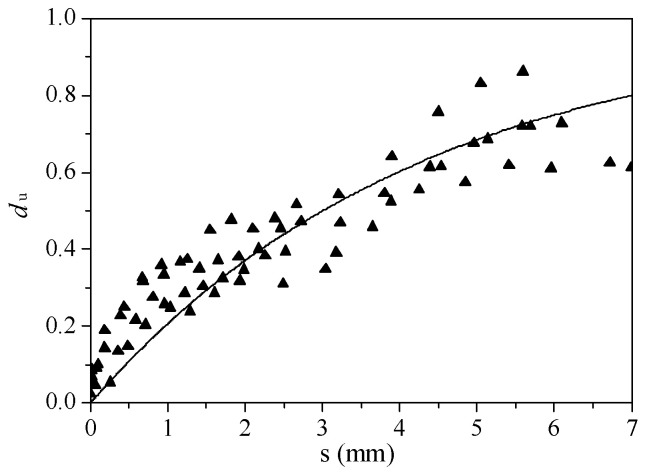
Relationship between damage factor and slip.

**Figure 12 materials-16-00252-f012:**
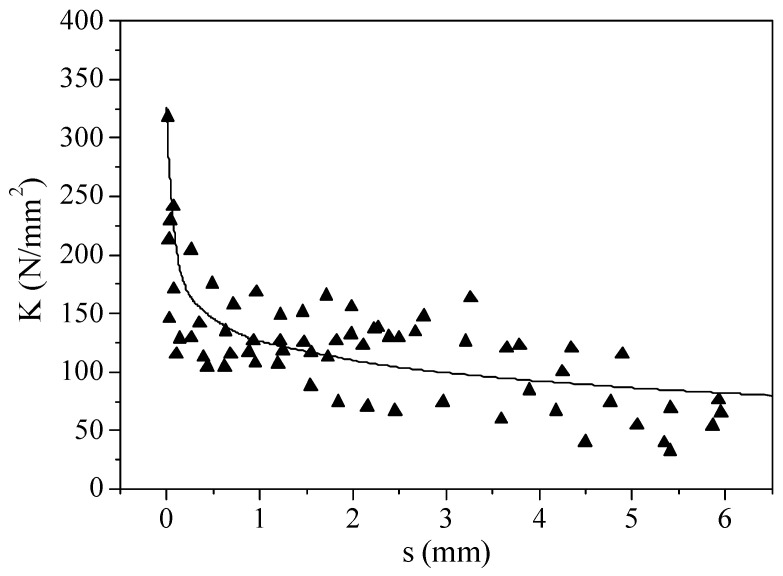
Relationship between unloading stiffness and slip.

**Figure 13 materials-16-00252-f013:**
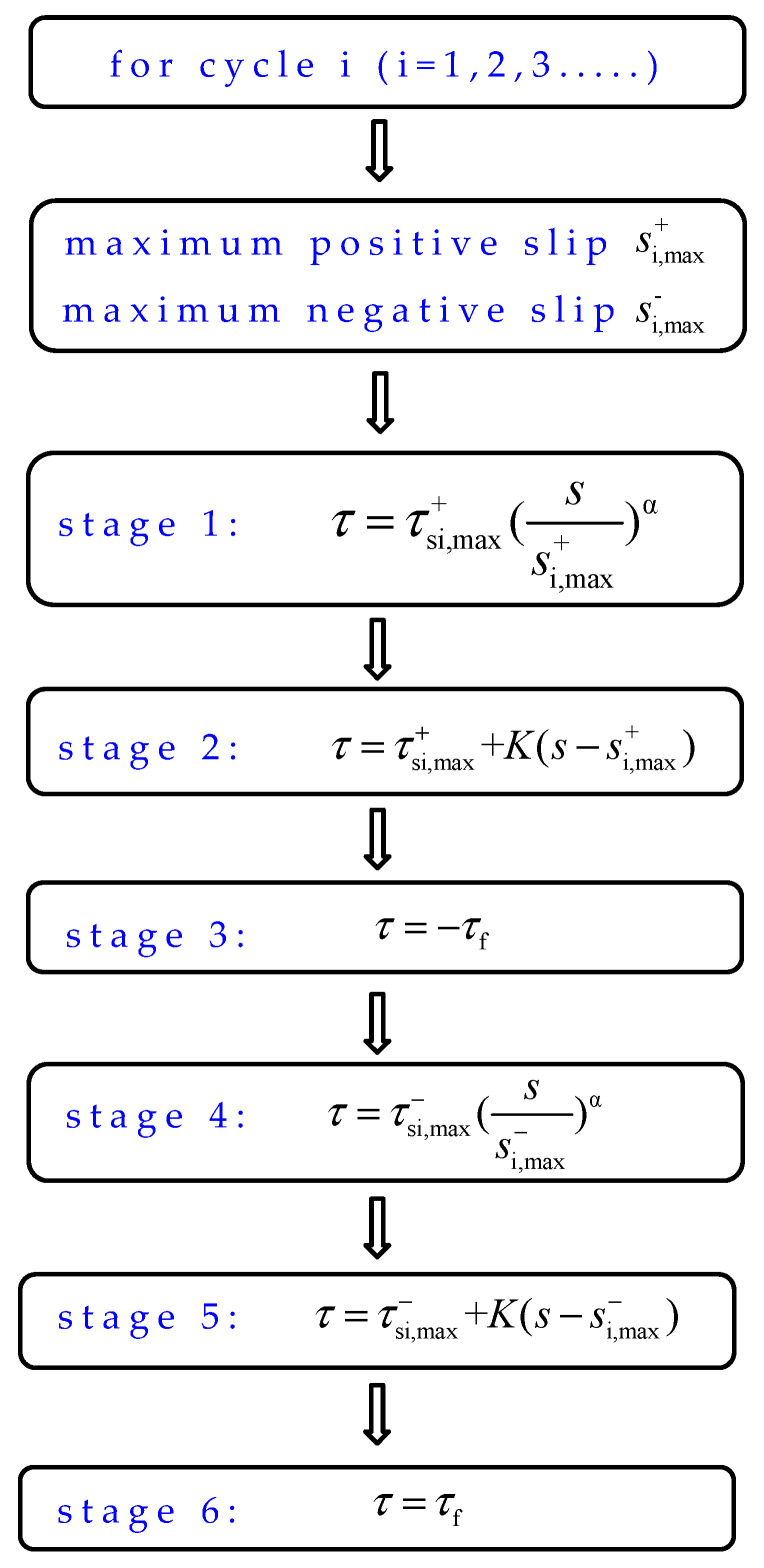
Calculation process.

**Figure 14 materials-16-00252-f014:**
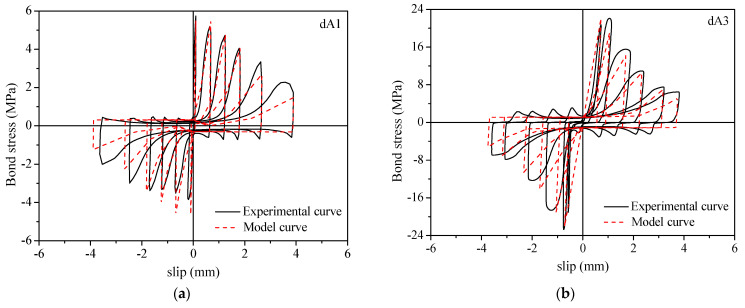
Comparison between the model curve and experimental curve under cyclic loading. (**a**) dA1; (**b**) dA3; (**c**) dB3; (**d**) dD2; (**e**) dD3; (**f**) dE3.

**Table 1 materials-16-00252-t001:** Mixture proportions and mechanical properties of ECCs.

Mixture	Binder				Water Binder Ratio	Sand Binder Ratio	Fibers/Vol.		*f*_cu_ (MPa)	*T* (kJ·m^−3^)	*f*_t_ (MPa)
	Cement	Fly Ash	Mineral Powder	Silica Fume			PVA	PE			
A1	45%	55%	-	-	0.31	0.36	1.5%	-	51.7	33.4	4.1
A2	50%	30%	20%	-	0.27	0.36	1.5%	-	71.0	26.8	3.7
A3	55%	20%	20%	5%	0.24	0.36	1.5%	-	81.3	28.7	5.1
B1	40%	55%	-	5%	0.29	0.36	1.5%	-	51.9	34.3	3.9
B2	40%	55%	-	5%	0.29	0.36	-	1.5%	51.8	105.5	3.1
B3	40%	55%	-	5%	0.29	0.36	-	2.0%	51.4	250.8	3.7

**Table 2 materials-16-00252-t002:** Specimen design and test results.

Code	ECC	*c*/*d*	*l*_a_/*d*	Cyclic Loading						Monotonic Loading			$\lambda_{u}$
				$\tau_{u}^{+}$ (MPa)	$\tau_{u}^{-}$ (MPa)	$\tau_{u}^{m}$ (MPa)	$\tau_{r}^{+}$ (MPa)	$\tau_{r}^{-}$ (MPa)	$\tau_{r}^{m}$ (MPa)	$\tau_{u}$ (MPa)	$\tau_{r}$ (MPa)	*k* _r_	
dA1	A1	3.3	5	4.188	−4.989	4.589	1.466	−0.827	1.146	6.879	2.700	0.414	0.667
dA2	A2	3.3	5	9.715	−16.427	13.071	4.361	−2.425	3.393	17.018	7.626	0.449	0.768
dA3	A3	3.3	5	20.680	−21.686	21.183	5.138	−4.019	4.579	28.231	6.991	0.246	0.750
dB1	B1	3.3	5	7.348	−8.431	7.889	2.407	−1.752	2.079	8.713	3.073	0.325	0.905
dB2	B2	3.3	5	9.480	−12.084	10.782	2.697	−2.565	2.631	11.976	4.072	0.366	0.900
dB3	B3	3.3	5	8.388	−9.683	9.036	2.294	−2.100	2.197	10.267	3.599	0.349	0.880
dD1	A1	2.5	5	4.064	−5.290	4.677	0.749	−0.656	0.702	7.402	1.456	0.196	0.632
dD2	A1	1.9	5	3.429	−3.716	3.572	0.892	−0.655	0.773	5.922	1.779	0.300	0.603
dD3	A1	1.4	5	3.626	−5.085	2.921	0.627	−0.428	0.527	4.860	1.153	0.237	0.601
dE1	A1	3.3	3	5.401	−5.797	5.599	1.999	−1.124	1.562	7.063	2.917	0.439	0.793
dE2	A1	3.3	4	4.729	−5.392	5.060	1.730	−1.366	1.548	5.805	2.156	0.372	0.872
dE3	A1	3.3	7	6.296	−8.034	7.165	0.978	−1.014	0.996	11.368	2.020	0.178	0.630

**Table 3 materials-16-00252-t003:** Ratio of negative peak bond stress to positive peak bond stress.

Parameter	dA1	dA2	dA3	dB1	dB2	dB3	dD1	dD2	dD3	dE1	dE2	dE3
$\tau_{s}^{-}/\tau_{s}^{+}$	0.839	0.591	0.954	0.872	0.784	0.866	0.768	0.923	0.713	0.932	0.877	0.784

## Data Availability

The data presented in this study are available on request from the corresponding author.

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
