# Peer review of "Bond Degradation Mechanism and Constitutive Relationship of Ribbed Steel Bars Embedded in Engineered Cementitious Composites under Cyclic Loading"

_materials, 2022, doi:10.3390/ma16010252_

Round 1

Reviewer 1 Report

1. There are significant differences in some of the mixture proportions and mechanical properties of ECC, as well as in specimen design and test results. The authors used three samples in each test group. Statistical validation of these data is needed.

2. Differences in mechanical properties should be explained, especially for flexural toughness T of Mixture B. The same comment concerns the tests results, which is observed for example in tu values ranged from7.063 MPa up to 11.368 MPa in specimens dE.

3. Degradation of ultimate bond strength is described by coefficient λu. This parametr must be described in detail, because provided description "The ratio of ultimate bond strength under cyclic loading to that under monotonic loading" is general. It would be valuable to include the precise formula of λu and discuss its effect on the ECC behaviour.

Author Response

Dear Reviewer,

Thank you for your comments concerning our manuscript entitled “Bond degradation mechanism and constitutive relationship of ribbed steel bar embedded in Engineered Cementitious Composites under cyclic loading”. Those comments are very valuable and great helpful for us to revise and improve our paper. We have substantially revised the manuscript and the language has been further polished. The revised portions are marked in red in the paper. We appreciate for editors’ warm work earnestly and hope that the correction will meet with approval. The main corrections in the paper and the responses to the reviewer’s comments are provided in attached file.

Reviewer 2 Report

The manuscript titled "Bond degradation mechanism and constitutive relationship of ribbed steel bar embedded in Engineered Cementitious Composites under cyclic loading" is about degradation mechanism between steel bar and ECC and affecting parameters. Finally, authors proposed constitutive relationship for stress-slip curves considering monotonic and cyclic loading. Paper is interesting but needs following issues to be considered:

* Abstract of paper lacks of logical order giving information about the results and proposal of relationship and should be re-written.

* Reviewer thinks that citation of references in the manuscript not complies with journal citation style.

* In the paper, authors used limited numbers of experimental results for the derivation of equations. Accordingly, proposed equations should have limitations. Authors should inform the practicioners about the limitations.

* Notations and units should be re-checked and corrected.

* There are missing details, structure of paper needs revision for clear understanding for the readers.

* Language lacks of grammar at times. Needs careful read and should be revised.

* There is a gap between construction of relations between parameters for the derivation of stress-slip curves. Authors should adequately inform readers about the parameters of the equation and relations.

* In section 5.3, or where appropriate, it is better to propose a flowchart (step by step procedure) for the calculation process of the curves for the ease of understanding and simple application for the readers.

* In general, language of the paper should be checked, some example are notated for the authors in the attached file.

* Additional explanations for the authors are also provided in attached file. Please read the file carefully and revise accordingly

Author Response

(The authors gave the same response as above.)

Round 2

Reviewer 1 Report

I accept the paper for publication.

Author Response

Thank you for your comments.

Reviewer 2 Report

The revised manuscript titled "Bond degradation mechanism and constitutive relationship of ribbed steel bar embedded in Engineered Cementitious Composites under cyclic loading" is improved in general. However, some technical issues not solved and should be clarified.

1) There should be erroneous in the correlation coefficient of Eq. 6. Authors should cliarify the calculation. See also comment 2.

2) In first revision, authors were recommended to change equation type Eq. 6. Because, equation did not fit well with real data as seen in Fig. 10. 

Authors are strongly recommended to try power function (y=ax^b). Accordingly, authors are referred to see comments of reviewer in the first revision that was given in the attached file. Approximate values for a, b values were also provided for authors.

3) Same issue is also valid for Eq. 8. See the comments that was given in the attached file in the first revision for detailed explaination.

4) Although authors plotted a Flowchart for the calculation process, it is superficial and does not give required information for the calculation process. So, Needs improvement. For example, it does not provide information about how to calculate max slip value for ithe cycles for both ascending and descending branch. How can reader calculate the maximum slip for each cycle (e.g., t+si,max & s+i,max )?

5) In Fig. 14, authors compared the model and experimental curve under cyclic loading. According to stress‐slip curves of specimens (dA3 and dB3) given in Fig. 3, cyclic curve of correspongding curves do not match in terms of number of cylcles and maximum slip values in horizontal axes. Needs clarification!

Author Response

Thanks for your comment. The specific response refers to the attachment.

Round 3

Reviewer 2 Report

"For the power function (y=ax^b), it can not reflect accurately the change trend of damage factor with the slip" is not true, but authors can advocate that "Eq. 6 is often used to describe the develop process of damage in existing references"

Accordingly, authors can add following sentence before Eq. 6 whixh can be done in proof reading process:

"Since Eq. 6 is often used to describe the develop process of damage in existing references [existing reference list], other functional forms can be also adapted [reference]"

existing reference list: list of some referencse that authors claim

reference: see the reviewer's recommended reference in first revision